# Evaluation of Long-Term Care Insurance Policy in Chinese Pilot Cities

**DOI:** 10.3390/ijerph16203826

**Published:** 2019-10-11

**Authors:** Yanzhe Zhang, Xiao Yu

**Affiliations:** Northeast Asia Study College, Jilin University, Changchun 2699, China

**Keywords:** evaluation, long-term care insurance, pilot cities, China

## Abstract

Since 2016, 15 pilot cities in China have implemented a long-term care insurance (LTCI) policy. The aim of this research was to explore the outcomes and evaluate the performance of the LTCI policy in the Chinese pilot cities and estimate the willingness of Chinese citizens to expand the formal implementation of LTCI policy in China. We gathered data from 1500 elderly people aged over 60 years in 15 pilot cities (100 surveys for each city) and the effective response rate was 77.8% (1167/1500). We relied on statistical analysis to elicit the outcomes and performance of LTCI implementation and an ordinal logit regression to analyze the factors associated with the extension of the LTCI policy. We examined factors associated with the perception according to sex, age, degree of disability, choices of care, living location, number of children, and monthly income. Among these factors, the relationship between living location and number of children of the family and the outcomes and performance of the LTCI policy in the pilot cities was significant. The rest of the factors showed no significance with the implementation of the LTCI in Chinese pilot cities. This study is among the first to explore the attitudes of Chinese citizens among those who have benefited from the LTCI policy in the pilot cities and contributes to identifying the outcomes of the LTCI in pilot cities to assist policymakers in their further implementation in China.

## 1. Introduction

Since 1963, mainland China has witnessed a rapid demographic shift [1]. In 2000, China had the largest population in the world, with a total population of over 1.3 billion, which reached 1.39 billion at the end of 2018 [2,3]. The problems caused by an aging society, such as providing services for the disabled and elderly care, have become massive challenges for the Chinese government. According to the National Bureau of Statistics of China, 241 million people were aged over 60 years in China at the end of 2017. This was an increase of over 10 million from 2016, accounting for 17.3% of the total population [4]. This makes China the only country in the world with an elderly population exceeding 200 million. China’s elderly population is estimated to reach approximately 490 million by the end of 2020; as Barlett and Phillips predicted, one in eight Chinese are likely to be aged over 70 years old by that point in time [5].

The World Health Organization (WHO) suggested that China initiate a long-term care insurance (LTCI) policy in their health care system to address the issue of providing care to disabled and elderly people [6,7,8,9]. To establish the LTCI policy and improve the development of the social security sector in China, many government sectors, such as the Ministry of Human Resources and Social Security (MOHRSS) and the Ministry of Civil Affairs (MOCA), have initiated the process, which has facilitated the reform of China’s health care system. The general office of the MOHRSS issued guidelines on implementing the LTCI policy in pilot cities (the 80th official document of the Chinese Central Government) on 27 June 2016 [10], confirming that China was officially launching a LTCI in 15 pilot cities, which included Shandong province, Jiangsu province, and Shanghai city (municipality directly under the Central Government). Those administrative regions issued and implemented regulations in their own areas [11,12].

The Chinese Government and MOHRSS are aiming to establish a fundamental LTCI policy framework in China in 2020. The LTCI is a social insurance system that provides basic life care and daily nursing services to the disabled and elderly. It is regarded as a “sixth insurance” in addition to social insurance provided for endowment, medical treatment, work-related injury, unemployment, and childbirth. According to the National Bureau of Statistic of China, China’s LTCI covered 57 million people, with 184,500 people receiving the benefits at the end of June 2018. On 9 April 2019, the state council announced that the LTCI pilot program should be further expanded by both the MOHRSS and National Health care Security Administration (NHSA) in China [10].

The LTCI policy recommendations include the following focal points: (1) LTCI principally covers urban citizens and those who live in rural areas; (2) LTCI mainly provides financial assistance to people who are disabled or elderly; (3) the LTCI participants are advised to pay 30% of the costs to meet the requirements of reimbursement; and (4) at this stage, pilot cities are encouraged to act on the unique policies in alliance with their own circumstances using best practices to expand LTCI to the national scale [12].

One of the challenges now faced by China’s health care system is whether the Chinese government should formally expand the implementation of the LTCI policy to the country scale. An evaluation of the outcomes and performance of the LTCI policy in pilot cities is crucial for government leaders and policymakers who are responsible for further decision making regarding the implementation of the LTCI policy in China.

Both quantitative and qualitative approaches were used to assess the evaluation of the LTCI policy in Chinese pilot cities. The quantitative approach was based on feedback from surveys that combine the issues of sex, age, degree of disability, choices of care, living location, number of children, and monthly income. The qualitative approach was built on answers in interviews with the political elites, including senior government officers, professors, and experts in the government. With regard to policy implementation and policy evaluation, the feedback from the surveys was dependent on the willingness of the Chinese citizens who participated and the answers from the political elite interviews were used to understand the choices in the health care system from a government perspective. Both will influence whether the LTCI policy should be further implemented in China as an official policy.

We aimed to empirically and prescriptively contribute to the contemporary studies of the LTCI policy in China. Few studies from this perspective on LTCI policy in China have been published, as most previous studies have been characterized by the financial functioning of the health care system and the Urban Employees Basic Medical Insurance that provides relief to the burden on both the individual families and governments in China [13]. This highlights the absence of investigations assessing the outcomes and performance of the LTCI policy as a pilot program in China. The existing approaches to studying the LTCI policy in China are limited by their narrow epistemological perspectives, as they focused on ideas, functioning, and the initiating process. Following Wang [14], we examined whether Chinese citizens and governments need to establish an LTCI policy in the health care system. The second main aim of this work was to examine whether the LTCI policy is promoting the development of the health care system in China. We contend that the LTCI policy has become a key policy instrument in the process of initiating a modern Chinese social security sector. The third and final contribution of this paper lies in the social security aspect of the health care system in China, which can hopefully guide Chinese policymakers to produce more progressive LTCI policy outcomes.

## 2. Materials and Methods

### 2.1. Data Collection

Due to the current implementation of the LTCI (Appendix A) policy in the pilot cities in China, the research on LTCI has not sufficiently explored the issues and factors that influence the decision to further officially implement LTCI at the country scale. Will the previous Family-Plan Policy affect the LTCI policy in the implementation process? The aim of this study was to fill these literature gaps.

In the winter of 2017, we started an investigation into evaluating the outcomes and performance of the LTCI policy in China. The aim of this investigation was to assess whether the citizens engaged in the implementation process were satisfied with the outcomes and performance of the pilot program. We associated with the Bureau of Human Resources and Social Security, Insurance Regulatory Bureau, Civil Affairs Bureau, among others, to conduct a survey that gathered data of LTCI participants from 15 pilot cities. A questionnaire entitled “An Assessment of Long-Term Care Insurance Policy” (Appendix B) was designed. The information collected in the survey included sex, age, degree of disability, the choices of care, living location, number of children, and monthly income. The feedback of the questionnaires was concluded into a 3-point scale: satisfied—1, neutral—2, and dissatisfied—3. The survey instrument was administered to 1500 individuals, who were covered by the LTCI from 15 pilot cities (100 participants in each city). According to the geographical distribution of pilot cities (3 cities in Western China, 5 cities in inner China, and 7 cities in coastal Eastern China), 300 of the 1500 participants were from Western China, 500 from inner China, and 700 from Eastern China. We received 1167 effective responses, for an effective response rate of 77.8% (1167/1500). Data from the remaining 333 participants were absent due to meaningless responses and missing records. The authors examined two research questions (RQs):RQ1. Are the citizens satisfied with the outcomes and performance of the LTCI policy in the pilot cities?RQ2. What is/are the main factors strongly associated with the implementation of the LTCI policy in the pilot cities?Given the results of the political elite interviews, we also conducted a third research question:RQ3. Are the political elites and policymakers aware of the importance of the LTCI policy?

### 2.2. Variables Definition

Table 1 shows that there were a total of 1167 effective responses to the questionnaire. The participants were enthusiastic and wanted to share their ideas about the LTCI policy. Of the effective feedback, 47.0% of respondents were men (*n* = 548) and 53.0% were women (*n* = 619).

A total of 29.8% of the effective participants (348 people) were aged from 60 to 69, 39.4% (460 people) were aged from 70 to 79, 23.0% (268 people) were aged from 80 and 89, and 6.6% were aged from 90 to 99 (77 people). We received 14 effective responses (1.2%) from people aged over 100 years old.

Among the survey responses, 15.2% of the effective participants (117 people) were not disabled at all, 19.5% were partly disabled (227 people), 41.8% were severely disabled (488 people), and 23.6% were completely disabled (275 people).

Of the respondents, 71.6% (835 people) preferred care at home. A total of 4.8% of the participants (56 people) preferred to pay the local communities for health care services and 12.1% (141 people) wanted to pay the nursing home. The remaining 135 people (11.6%) were paying the hospital to look after them.

We generated data from 15 pilot cities, which we placed into three categories: western cities (Shihezi, Chongqing, and Chengdu), inner China cities (Changchun, Qiqihaer, Anqing, Shangrao, and Jinmen), and eastern coast cities (Chengde, Shanghai, Nantong, Suzhou, Ningbo, Qingdao, and Guangzhou). A total of 207 (17.7%) effective responses were received from the western cities, 366 (31.4%) from the inner China cities, and 594 (50.9%) from the eastern coast cities.

According to the one-child policy, most Chinese families should only have one child. However, a few families have no descendants or more than one child. Therefore, 81.2% (947 people) of the total effective responses were from one-child families, 8.7% (102 people) from families with two or more children, and 10.1% (118 people) were received from childless families.

The monthly incomes of the participants in the survey were divided into six groups: 0–999 RMB (17.0%), 1000–1999 RMB (23.8%), 2000–2999 RMB (17.6%), 3000–4999 RMB (18.0%), 5000–7999 RMB (15.6%), and over 8000 RMB (8.1%).

### 2.3. Variable Analysis

We used descriptive statistics to study the population. Satisfaction with LTCI was set as the dependent variable and perceptions based on sex, age, degree of disability, choices of care, living location, number of children, and monthly income were individually recognized as independent variables. The relationship between satisfaction with the LTCI policy and the personal status of policy beneficiaries, such as sex, age, degree of disability, choices of care, living location, number of children, and monthly income (*n* = 1167), was analyzed. These were analyzed using an ordered logit regression because the dependent variables were categorized and ordered. In the model, sex, choices of care, and living location were treated as the unordered categorical variables. Age, monthly income, degree of disability, and number of children were treated as the categorical variables. All the statistical analyses were two-sided and performed using SPSS (17.0 Version SPSS Inc., Chicago, IL, USA) and STATA (14 MAC Version Stata Corp, College Station, TX, USA), and the statistical significance was set to *p* < 0.05.

## 3. Results

### 3.1. Sample Characteristics

Among the 1167 effective responses, 72% (840 people) were satisfied with the outcomes and performance of the LTCI policy in the pilot cities. A total of 8% (93 people) expressed a neutral attitude toward the LTCI policy and 20% (234 people) were displeased with LTCI as a pilot policy.

### 3.2. Quantitative Analysis

Table 2 is the result of ordered logistic regression analysis, including coefficient, standard deviation, p value, confidence interval, etc. It shows the basic characteristics of the distribution of effective responses to the “An Assessment of Long-Term Care Insurance Policy” survey. First, we employed model fitting information, and the result showed that *p* Significance (Sig.) was 0.000, indicating the overall significance of the model. Second, a test of parallel lines was conducted and the *p* (Sig.) value was 0.349, which is greater than 0.05, which satisfies the test of parallel lines. This indicates that the parametric estimated value of the ordered logistic regression model is reliable and reflects the influence of independent variables in the model on the dependent variable.

Significant relationships between living location and number of children of the family and the outcomes and performance of the LTCI policy in the pilot cities were shown in Table 2. These implied that the majority of the population movement and migration was from the western, landlocked areas to the east, coastal cities, it is difficult for an individual to deal with massive, detailed and time-consuming nursing tasks at home. Thus, the LTCI plays an important role in undertaking responsibility in safeguarding the legitimate rights and interests, maintaining their basic survival, providing material and spiritual support and etc., of the elderly. The issues of sex, age, degree of disability, choices of care, and monthly income are also revealed in Table 2; the results were not significant (*p* > 0.05).

Moreover, people living in the western cities reported higher levels of satisfaction than those living on the east coast. People without children were more satisfied than those with two or more children, people with one child were more satisfied than people with two or more children, and people without children were more satisfied than people with one child.

### 3.3. Qualitative Analysis

We conducted interviews with five elite politicians. Two of them were interviews with professors in government studies at Chinese universities and governmental research institutes. Professor Peng Liu is an expert in demographic studies working at the Party School of the Central Committee of C.P.C.; Professor Yuetong Xu is a specialist in policy studies at the China Academy of Governance. In addition, three senior officers at each level of Chinese government were interviewed. One of the authors conducted personal interviews with them at their working places. The questions were relevant to the policy-oriented learning framework and China’s social security sector. The duration of the interviews spanned around 1–2 h. The author was allowed to take notes in English.

## 4. Discussion

The formal implementation of the LTCI policy is at the center of the current reform in China’s health care system [15]. The LTCI policy has been recognized as a rational policy instrument that can provide a function of financial supplement to aid the development of social security in China [16]. An increasing number of researchers have begun to examine the outcomes and performance of the LTCI policy in pilot cities in China, and no research has assessed what factors are influencing the implementation of the LTCI policy [17]. We are amongst the first to evaluate the factors that are significantly associated with further implementation of the LTCI policy among this population and the findings of this paper offer important insights into political elites and policymakers in bettering the health care system in China.

### 4.1. The Legacy of the Family-Plan Policy and the Long-Term Care Policy

The major result of the Family-Plan Policy was to increase the age of marriage and childbirth [18,19]. This has maintained the population at 1.3 billion, which is estimated to be 300 million smaller by the end of 2010 than it would have been if the one-child policy had not been implemented [20,21]. In China, despite the evident sacrifice of individual rights in the interest of the collective rights of Chinese society, the policy has generally been viewed as the best strategy for controlling the population explosion. However, the policy has led to an irremediable risk for social security, as, by 2050, China will be a gray society [5,22]. This has placed Chinese society and its young at serious risk, especially in rural areas, where the elderly heavily rely on their families for care. More recently, theoretical research and practical explorations have been conducted to manage the aging society in China, especially in the field of social security.

One of our major findings was that the number of children in a family is the key factor influencing the further implementation of LTCI in China. Since the People’s Republic of China was founded, the political elites and policymakers have realized that overpopulation and a surplus of labor were leading to some serious social problems, such as unemployment, a shortage of food and clothes, an absence of public health and educational resources, and the underdevelopment of the economy [23]. Therefore, the Deng Xiaoping administration initiated the Family-Plan Policy, the one-child policy, which mandated that a couple should only have one child in each family [24]. The aim of this policy was to limit the family size to three people and restrict the population to below 1.3 billion by the end of the 20th century [19,25].

Consequently, the one-child policy considerably improved the economic situation and alleviated the incongruity between the population and public resources of China in the short term. However, the negative consequences of the one-child policy began to accumulate, having long-term adverse impacts on Chinese society. One of the main negative consequences of this policy is the aging society. Nearly 40 million disabled and partially disabled elderly people reside in China, including nearly 10 million completely disabled elderly people in 2017 [26,27]. The problem with an aging society and the deficiency in LTCI for disabled people is resulting in an irremediable problem for social security in China as, between 2022 and 2035, China will face a serious situation caused by increasing numbers of elderly people, with an annual net increase of 11.52 million and a 3.41% annual growth rate. The total number of elderly people will reach 420 million by year 2035 [5,28,29,30,31,32,33].

To sum up, the sharp decrease of the number of children and the significant increase of the number of only children in the family results in the traditional nursing model of multiple children taking care of elderly parents suffer a serious impact. The provision of care within the traditional family is gradually being lost, and the nursing needs of the elderly cannot be fully met. Studying, working and moving away from home will lead to a phenomenon of empty nest families. If the only child is disabled or dies, his or her parents will be trapped in a situation in which they have no one to take care of them. As the first generation of parents in the only child family have now reached old age, the risks of long-term care faced by these elderly parents are increasingly prominent. While these families are in trouble, they also affect the stability and development of the society.

Since the implementation of the family planning policy, parents of the only child family have actively complied with the government requirement and contributed to the population control of the Chinese society. Therefore, the Chinese government and society should initiate a policy product in response to the long-term care problem of Chinese families. At present, the focus of LTCI is how to deal with the risks of the one child family. The lessons and practice in various countries have proven that the LTCI is one of the most effective policy tools to deal with the risks of providing care to the elderly. It can reduce the financial burden of the elderly while providing nursing services.

### 4.2. Cultural Shifts and the Long-Term Care Policy

China has a 5000-year history with 56 nations and national ethnicities. The notion of filial obedience plays a decisive role in Chinese culture and norms, and even affects the current Chinese governance and ethnic code [34,35]. Filial obedience is the center of Confucianism and has been recognized as an ethical law of the Chinese for over 2000 years. It requires the Chinese to follow the ethical code of filial obedience to administrate their families and behaviors [36]. In line with this code, providing care to the disabled and elderly is the priority of the children in the family. In turn, China as a nation, relies heavily on the private care of the elderly at home [37]. With the founding of the People’s Republic of China, the welfare entitlement of laborers was offered by the work units in China [38,39]. Due to the insolvencies of state-owned enterprises and widespread layoffs of labor workers in 1998, “optimizing the allocation of labor” has been a goal of the central government in China [40]. The welfare of Chinese laborers was provided by the MOHRSS. Currently, nearly 90% of Chinese citizens think that the government and society as a whole should provide LTCI services to disabled and elderly people in China [41]. Zhang stated that “[W]ith the succession of China’s aging problem, the disabled and elderly population with a need for long-term care has significantly increased. Therefore, it is very important to initiate a LTCI policy and mechanism that is in accordance with the China’s circumstance”. This is a significant cultural shift in administration both in the norms and values of Chinese society.

Since the New Millennium, children have not been able to look after their parents and parents-in-law as they had previously. There are many complex reasons for this; two of the most important issues are the rural-to-urban and small-city-to-metropolis migrations [42]. As a consequence, children are not able to provide care for their elderly in rural areas and small cities due to the distance between them and their families. As Zhang Tiemin stated,
“[N]owadays [t]here [are …] a large number of elderly people [who] cannot be provided [with] care[…] by their children because a large proportion of their children are far away from them as a consequence of work, immigration, or marriage”.[43]

The other finding in this research was that people from inner China cities (Anqing, Changchun, Chengde, Chengdu, Jingmen, Qiqihaer, Shangrao, and Shihezi) expressed more approval with the LTCI compared with people who lived in the cities in Eastern China (Guangzhou, Nantong, Ningbo, Suzhou, and Shanghan). Due to the urban and economic development in China, many laborers are migrating from rural areas to cities. Simultaneously, many people from small- and medium-sized cities are moving to metropolises [37,42,44,45]. According to the report on population mobility in China by the National Health and Family Planning Commission of the People’s Republic of China (NHFPC), the number of people in China moving from towns to cities increased from 230 million in 2011 to 245 million in 2016 [42]. As Zhang observed, 44% of the Chinese population lived in urban areas in 2005, which was predicted to reach 66% by the end of 2025 [20], meaning that one in every six Chinese people migrated [46]. The elderly in western cities and inner China usually preferred to stay individually in their hometown city without support from their children [35]. This was the main reason for them to purchase LTCI. The parents who were living in coastal cities were usually provided care by their children living nearby. They had some desire to buy LTCI, but not as much as the people living in inner China.

Another practical implication in this study is that sex, age, degree of disability, choices of care, and monthly income had no significant relationship with their desire to purchase LTCI. For example, China has no existing standard to define the degree of disability [15,39]. LTCI can serve as a whole health care system mechanism that stimulates the development of China’s social security sector. However, setting up criteria for the degree of disability, as achieved by Germany and Japan in their health care systems, is too difficult, because the regional character is complex in China compared to the rest of the world. Just as Zhang addressed that China is a big country with huge differences in cultures and development paces among regions. The promotion of LTCI policy should focus on the differentiation in developing the local social security system [41].

Moreover, monthly income was not strongly associated with the implementation of the LTCI policy in the pilot cities. China’s per capita income was 28,228 RMB in 2018 [47]. The cost of LTCI in the pilot cities was approximately 1600 RMB annually, which is only 6% of Chinese per capita income [48]. With the economic development and the promotion of knowledge about LCTI, the feedback from the participants showed a willingness to pay for the services rather than save money for future distress.

LTCI can effectively relieve the burden of social hospitalization. Once hospitalized for long-term care, the elderly, rich or poor, all live beyond their incomes and struggle to return to better health. Xu stated:
[I]n fact, most of the diseases do not need to be treated in hospitals. If timely prevention and proper long-term care services are provided, people need not be hospitalized. Only 10% of the modern diseases need to be treated in the hospital. If home care and community care are developed rapidly based on the support of LTCI, the phenomenon of “bed occupation” in hospitals for the elderly with chronic diseases or disability will be reduced significantly.[48]

LTCI will also relieve the pressure of medical insurance fund payments and the social problem of the difficulty in seeing a doctor. LTCI is conducive to China’s economic growth and employment. China’s LTCI market will become the next new economic growth point of China’s new normal development. With the increase in the proportion of the elderly population, the elderly service industry will provide 11.55 million jobs in the next 5 to 10 years, which is a good opportunity to promote the employment of older women in urban and rural areas and college students majoring in nursing, psychology, social work, human resources, and other related majors, and expand their management positions in China [36].

The long-term care policy was introduced and has taken strong steps in China. It has also led to the betterment of China’s social security. The analysis of the elites provided in the previous sections presents a detailed and exploratory study of how the political elites and policy makers are aware of the importance of the LTCI policy. The results presented here suggest that the LTCI policy should expand to the national scales in China. China’s aging problem is a legacy of the Family-Plan Policy that has been implemented since 1979, the new demographic issues from the New Millennium, and the culture shifts of Chinese society today. The LTCI policy is a policy tool to deal with these social problems. In addition, China possesses a combination of some of the richest and poorest areas in the world, hence, issues such as the capacity to pay for social insurance are fundamental to be addressed both timely and appropriately. Obviously, from the results of the pilot program, the influence of LTCI insurance has been effective. There were 38 million people who had signed up for the LTCI by the end of the third-quarter of 2017. Prime Minister Li Keqiang said that the LTCI will be expanded by Chinese government in order to provide happy twilight years to each elderly person in China [41].

Some limitations exist in our methodology that should be addressed. First, this research was mainly based on a quantitative method. The data were gathered randomly through surveys from 15 pilot cities. We did not segment the respondents by educational background, social status, etc. The feedback from the surveys may not be accurate or reflective. Further studies are needed to identify and quantify more factors. Second, this paper was a part of research on the policy evaluation of China’s LTCI. This policy was influenced by the German health care system [7,8,49]. The compatibility between the Chinese government and Germany has not been emphasized in this paper. This issue needs to be acknowledged in further studies. Third, some differences exist in LTCI among the 15 pilot cities in the policy implementation processes due to their local circumstances. In administrative science, we usually call these policy gaps. The policy gaps among the pilot cities should be explored in future studies. Fifth and lastly, this study contained a few elite interviews. The political elites were from both the central government and local governments, but did not cover all the pilot cities. Further studies are needed to obtain more details from the political elites from each pilot city.

## 5. Conclusions

The aim of the LTCI policy is to relieve the problem created by an aging society and the relevant issues with the health care system in China. As mentioned in Section 2, we addressed three research questions in this paper: Are the citizens satisfied with the outcomes and performance of the LTCI policy in the pilot cities? What is/are the main factors that are strongly associated with the implementation of the LTCI policy in the pilot cities? Are the political elites and policymakers aware of the importance of the LTCI policy? To answer the first question, we used descriptive statistics to determine the number of people who are satisfied with the outcomes and performance of the LTCI policy in the pilot cities. Although some feedback suggested that they were dissatisfied or held a neutral attitude toward the policy, the satisfaction rate was 72.24%.

To answer research question two, we employed ordered logistic regression to analyze the data, which we generated from an administered survey named “An Assessment of Long-Term Care Insurance Policy”. We found that living location and number of children were significantly associated with the further implementation of the LTCI policy in China. By addressing the third research question, we conducted a series of interviews with senior officers, policymakers, and professionals. The answers were positive and constructive with regard to the current status of the health care system and social security in China.

In summary, the findings of this study provide three contributions. First, we explored the satisfaction rate of the LTCI policy in the pilot cities. Second, we assessed the outcomes and performance of the LTCI policy and found that living location and number of children were significantly associated with the further implementation of LTCI. Third, we observed analytical and methodological variables that can be used to supervise the further implementation of the LTCI policy in China. This study is the first comprehensive study of the LTCI policy and the significantly associated factors in China. Our findings have crucial implications for an aging society, social sector reform, and policy evaluation literature, as the LTCI policy aims to provide relief to the aging society in the Chinese health care system.

## Figures and Tables

**Table 1 ijerph-16-03826-t001:** Definitions and descriptions of variables included in the survey (*n* = 1167).

Variable	Measurement	Min.	Max.	Percentage (*N*)
Sex	1 = Male	1	2	47.0% (548)
	2 = Female			53.0% (619)
Age	1 = 60–69	1	5	29.8% (348)
	2 = 70–79			39.4% (460)
	3 = 80–89			23.0% (268)
	4 = 90–99			6.6% (77)
	5 = 100 and Above			1.2% (14)
Degree of Disability	1 = Not At All	1	4	15.2% (177)
	2 = Partly Disable			19.5% (227)
	3 = Severe Disable			41.8% (488)
	4 = Completely Disable			23.6% (275)
Choices of Care	1 = At Home	1	4	71.6% (835)
	2 = Local Community			4.8% (56)
	3 = Nursing House			12.1% (141)
	4 = Hospitals			11.6% (135)
Living Location	1 = Western Cities	1	3	17.7% (207)
	2 = Inner China Cities			31.4% (366)
	3 = Eastern Coast Cities			50.9% (594)
Number of Children	1 = 0 child	1	3	10.1% (118)
	2 = 1 child			81.2% (947)
	3 = 2+ children			8.7% (102)
Monthly Income	1 = 0–999 RMB	1	6	17.0% (198)
	2 = 1000–1999 RMB			23.8% (278)
	3 = 2000–2999 RMB			17.6% (205)
	4 = 3000–4999 RMB			18.0% (210)
	5 = 5000–7999 RMB			15.6% (182)
	6 = 8000 RMB and Over			8.1% (94)
Degree of Satisfaction	1 = Satisfied	1	3	72.2% (843)
	2 = Neutral			7.9% (92)
	3 = Dissatisfied			19.9% (232)

**Table 2 ijerph-16-03826-t002:** Coefficient of ordinal logit regression (*n* = 1167).

Variables	Satisfied (%)	Neutral (%)	Dissatisfied (%)	Coefficient (S.E.)	*p*-Value	95% CI
Gender Male	391 (71.3%)	41 (7.5%)	116 (21.2%)	0.165 (0.136)	0.222	−0.100–0.431
Female	452 (73.0%)	51 (8.3%)	116 (18.7%)			
Age 60–70	259 (74.4%)	23 (6.6%)	66 (18.0%)	0.050 (0.630)	0.937	−1.186–1.286
70–80	332 (72.2%)	34 (7.4%)	94 (20.4%)	0.117 (0.627)	0.851	−1.000–1.345
80–90	183 (68.3%)	28 (10.4%)	57 (21.3%)	0.302 (0.632)	0.633	−0.936–1.540
90–100	60 (77.9%)	2 (2.6%)	15 (19.5%)	−0.141 (0.676)	0.834	−1.466–1.183
100+	9 (64.3%)	5 (35.7%)	0 (0.0%)			
Degree of Disability Not At All	124 (70.1%)	15 (8.5%)	38 (21.4%)	0.298 (0.224)	0.184	−0.142–0.738
Partly Disable	163 (71.8%)	17 (7.5%)	47 (20.7%)	0.193 (0.217)	0.374	−0.233–0.619
Severe Disable	346 (70.9%)	39 (8.0%)	103 (21.1%)	0.214 (0.186)	0.251	−0.151–0.579
Completed Disable	210 (76.4%)	21 (7.6%)	44 (16.0%)			
Choice of Care Home	595 (71.3%)	72 (8.6%)	168 (20.1%)	0.389 (0.241)	0.107	−0.083–0.861
Local Community	41 (73.2%)	2 (3.6%)	13 (23.2%)	0.300 (0.394)	0.447	−0.472–1.072
Nursing House	100 (70.9%)	9 (6.4%)	32 (22.7%)	0.412 (0.300)	0.169	−0.176–1.000
Hospitals	107 (79.2%)	9 (6.7%)	19 (14.1%)			
Monthly Income 0–999 RMB	140 (70.7%)	16 (8.1%)	42 (21.2%)	0.461 (0.309)	0.135	−0.144–1.066
1000–1999 RMB	195 (70.1%)	26 (9.4%)	57 (20.5%)	0.429 (0.297)	0.149	−0.154–1.011
2000–2999 RMB	150 (73.2%)	13 (6.3%)	42 (20.5%)	0.344 (0.308)	0.265	−0.261–0.948
3000–4999 RMB	143 (68.1%)	19 (9.1%)	48 (22.8%)	0.593 (0.305)	0.052	−0.004–1.190
5000–7999 RMB	140 (92.1%)	10 (6.6%)	2 (1.3%)	0.129 (0.318)	0.685	−0.494–0.752
8000 RMB and Above	75 (79.8%)	8 (8.5%)	11 (11.7%)			
Living Area Western Cities	169 (81.6%)	9 (4.4%)	29 (14.0%)	−0.654 *** (0.203)	0.001	−1.052–−0.257
Inner China Cities	262 (71.6%)	24 (6.5%)	80 (21.9%)	−0.083 (0.152)	0.588	−0.381–0.216
Eastern Coast Cities	412 (69.4%)	59 (9.9%)	123 (20.7%)			
Number of Children 0 child	78 (66.1%)	10 (8.5%)	30 (25.4%)	−0.603 ** (0.279)	0.03	−1.149–−0.057
1 child	714 (75.4%)	69 (7.3%)	164 (17.3%)	−1.064 *** (0.211)	0	−1.477–−0.651
2+ children	51 (50.0%)	13 (12.8%)	38 (37.2%)			
Observations	*N* = 1167					
Degree of freedom	20					

Note: *** *p* < 0.01 and ** *p* < 0.05.

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
