# Peer review of "Evaluation of Long-Term Care Insurance Policy in Chinese Pilot Cities"

_ijerph, 2019, doi:10.3390/ijerph16203826_

Round 1

Reviewer 1 Report

I believe that the revised version of the paper shows substantial improvements over the initial submission, and most of my comments have been adequately addressed. I appreciate the efforts.

Author Response

Dear Reviewer:

Thank you very much indeed for you kindly responses. Those truly encourage me to my future research and studies. Your help is really appreciated.

Yours faithfully

Reviewer 2 Report

There are visible changes in the manuscript from the original version, but there is an incoherent change in Table 2. Before, all the variables involved in the ordinal logistic regression model were assigned by the individual coefficient and its related p-value. Now, all the levels of the variables have their own figures. Is it done in accordance with the methodology described in the paper?

In the 3.3 chapter, you mention a qualitative analysis. It is very difficult to find outcome in the text. Only the third research question is visible. Where are the results? Only a sole sentence is mentioned in the Conclusion chapter.

The regression outcome is described very weakly. Much more space should be devoted to description of the regression model.

In Table 2, why there is regression coefficient at a level of zero? I presume, it is because of the referred level of the variable. But, this fact should be mentioned in the methodology. It is very important in a process of interpretation successively, which level odds are computed to.

Also, the text should be proofread. There are still grammar mistakes and formal mistakes – for instance:

– “policymakers” instead of “policy makers”;

– in the reference 6: “Blondeau. J” instead of “Blondeau, J.”;

– several references are cited in a wrong way with round parentheses instead of square parentheses;

– in the 3.2 chapter, “p Significant” is mentioned instead of “significance” or “statistical significance”.

Author Response

Dear Reviewer:

Thank you very much for you comments. It is really helpful to improve the readership of my paper. I have done the revision as followings and hopefully they would satisfy your comments.

Response to the first question:

The previous version was not rigorous. There was only one P-value as some variables were treated as the continuous variables. But now, these variables are treated as the categorical variables. That is why all levels of the variables have their own figures. Moreover, this has no problems in accordance with the methodology described in the paper.   

Response to the second question:

The part 4 (discussion part) is the combination of results to both quantitative analysis and qualitative analysis. According to your kindly comments, there are three paragraphs have been added to make the result stronger to compare with the previous version.

Response to the third question:

The description of regression outcome has been rewrote and enriched (from line 208 – line 229).  

Response to the fourth question:

The parameter of 0 is truly redundant and useless to this article. I deleted them. 

Response to the fifth question:

This paper has been proofread by the English experts of MDPI again. ‘Policymakers’ has replaced the ‘Policy makers’. The reference 6 has been corrected. The round parentheses have been changed into square parentheses. The “p Significant” is now the “significance” in the 3.2 chapter.

Thank you very much indeed again about commenting my paper. Your comments will be highly appreciated and enlighten my future studies.

Best wishes,

Reviewer 3 Report

China has the largest population of elderly in the world, which poses a challenge to the delivery of health care services to this population. LTCI is a potential policy tool to ensure the disabled and elderly have adequate access to the Chinese health care system. The researchers investigate the impact of LTCI policy using a survey instrument and explore attitudes of Chinese citizens among those who benefited from the LTCI policy.

This study has important policy value, as it seeks to understand how LTCI can improve the lives vis-a-vis improved access to health care of a rapidly aging Chinese population. However, despite the importance of this study, there are several changes that should be made to improve the quality of this manuscript

lines 107: The language should be changed to "the survey instrument was administered to 1500 individuals." Not all individuals that received the survey responded.

lines 165: Stata Corp is located in College Station, Texas, USA.

lines 192-195: Discussion of the qualitative results in this section is sparse. It may be useful to mode some of the discussion that describes the qualitative results from the "DISCUSSION" section to the "QUALITATIVE ANALYSIS" section. Alternatively, an additional sentence could be added to the qualitative analysis section to inform the readers that the authors are going to discuss the qualitative results in the discussion.

Author Response

Dear Reviewer:

Thank you very much for you comments. It is really helpful to improve the readership of my paper. I have done the revision as followings and hopefully they would satisfy your comments.

Response to the first question:

The sentence has changed to “The survey instrument was administered to 1500 individuals, who were covered by the LTCI from 15 pilot cities (100 participants in each city).” According to your comments in line 124 in this version.

Response to the second question:

It is now “STATA (14 MAC Version Stata Corp, College Station, Texas, USA),” in line 198 in this version.

Response to the third question:

The discussion of the qualitative results has been enriched from line 269 – line 278 in Chapter 3.3. And the further brief discussions have been added from line 328 to line 344; line 407 – line 410; line 444 – line 457 at part 4 in this version.

Moreover, This paper has been proofread by the English experts of MDPI again.

Thank you very much indeed again about commenting my paper. Your comments will be highly appreciated and enlighten my future studies.

Best wishes,
